# Proteomics for Biomarker Discovery for Diagnosis and Prognosis of Kidney Transplantation Rejection

**DOI:** 10.3390/proteomes10030024

**Published:** 2022-07-02

**Authors:** Luís M. Ramalhete, Rúben Araújo, Aníbal Ferreira, Cecília R. C. Calado

**Affiliations:** 1Blood and Transplantation Center of Lisbon, Instituto Português do Sangue e da Transplantação, Alameda das Linhas de Torres, n° 117, 1769-001 Lisbon, Portugal; luis.m.ramalhete@edu.nms.unl.pt; 2NOVA Medical School, Faculdade de Ciências Médicas, Universidade NOVA de Lisboa, 1169-056 Lisbon, Portugal; rubenalexandredinisaraujo@gmail.com (R.A.); anibal.ferreira@nms.unl.pt (A.F.); 3Centro Hospitalar de Lisboa Central, Serviço de Nefrologia, 1050-099 Lisbon, Portugal; 4Nova Medical School, Hospital Curry Cabral, Serviço de Nefrologia, 1050-099 Lisbon, Portugal; 5ISEL-Instituto Superior de Engenharia de Lisboa, Instituto Politécnico de Lisboa, R. Conselheiro Emídio Navarro 1, 1959-007 Lisbon, Portugal; 6CIMOSM—Centro de Investigação em Modelação e Otimização de Sistemas Multifuncionais, ISEL, 1959-007 Lisbon, Portugal

**Keywords:** kidney allograft, proteomics, biomarker, rejection, biofluids, exosomes

## Abstract

Renal transplantation is currently the treatment of choice for end-stage kidney disease, enabling a quality of life superior to dialysis. Despite this, all transplanted patients are at risk of allograft rejection processes. The gold-standard diagnosis of graft rejection, based on histological analysis of kidney biopsy, is prone to sampling errors and carries high costs and risks associated with such invasive procedures. Furthermore, the routine clinical monitoring, based on urine volume, proteinuria, and serum creatinine, usually only detects alterations after graft histologic damage and does not differentiate between the diverse etiologies. Therefore, there is an urgent need for new biomarkers enabling to predict, with high sensitivity and specificity, the rejection processes and the underlying mechanisms obtained from minimally invasive procedures to be implemented in routine clinical surveillance. These new biomarkers should also detect the rejection processes as early as possible, ideally before the 78 clinical outputs, while enabling balanced immunotherapy in order to minimize rejections and reducing the high toxicities associated with these drugs. Proteomics of biofluids, collected through non-invasive or minimally invasive analysis, e.g., blood or urine, present inherent characteristics that may provide biomarker candidates. The current manuscript reviews biofluids proteomics toward biomarkers discovery that specifically identify subclinical, acute, and chronic immune rejection processes while allowing for the discrimination between cell-mediated or antibody-mediated processes. In time, these biomarkers will lead to patient risk stratification, monitoring, and personalized and more efficient immunotherapies toward higher graft survival and patient quality of life.

## 1. Introduction

Renal transplantation is the treatment of choice for end-stage kidney disease as it enables a much higher quality of life than dialysis. However, allografts can be lost due to several causes, ranging from events related to the pre-transplant period (e.g., preformed antibodies directed to the donor, pre-transplant organ stress, mismatch between recipient and donor), the operative period or due to infections, graft immune rejection and side effects of the immunosuppressive drugs therapy [1,2,3,4,5].

The graft immune rejection can be clinically classified as hyperacute, acute (AR), and chronic rejection (CR). Hyperacute rejections are rare events, occurring within minutes to a few hours after transplantation, and result from the interaction between existent preformed antibodies in the recipient, with graft antigens present in high density at the graft vascular endothelium. These antibodies can be isoagglutinins (blood group-related antibodies) and/or antibodies directed to the major histocompatibility complex (MHC), also known as human leukocyte antigens (HLA). Anti-HLA antibodies can be present in the recipient due to previous sensitization events such as pregnancies, blood transfusions, or even due to a previous transplant [6,7,8,9,10].

CR is often more difficult to diagnose since it is characterized by a progressive decline of renal function that is also characteristic of late acute rejection episodes and drug nephrotoxicity, among other renal diseases [6,11,12]. AR usually occurs within the first 6 months and is caused by infiltrating immune cells of the recipient on the graft organ due to cellular rejection (known as T-cells-mediated rejection (TCMR)) or by the presence of antibodies (known as antibody-mediated rejection (ABMR)), or by both mechanisms [6,13,14,15,16].

AR is a multistep process that starts with an immune system activation, followed by an inflammation process and possible organ tubule interstitial injury, resulting in organ damage or recovery. Besides strongly promoting the subsequent CR, AR still accounts for up to 12% of all graft losses [17,18]. Usually, early biopsies (before 1 year) present a treatable TCMR condition, while late biopsies often present an ABMR and are strongly associated with severe graft rejection [13,19,20,21,22]. TCMR events have been significantly reduced with the improvement in immunosuppressive agents in the early 1980s by cyclosporine and in the 1990s by tacrolimus, mycophenolate mofetil, and more recently with the introduction of lymphocyte depleting agents [23]. In the case of ABMR, the emergence of anti-HLA antibodies, either donor-specific antibody (DSA) or non-donor-specific DSA (non-DSA), are associated with poor kidney allograft outcomes, with special emphasis on DSA, as they have a higher impact on graft survival than non-DSA [24,25,26]. The mechanism associated with ABMR is highly complex and may occur either via complement-dependent cytotoxicity or complement-independent pathways of antibody-mediated cellular cytotoxicity [27,28].

Whereas clinical decisions based on conventional analysis of, e.g., serum creatinine, urine volume, and proteinuria may alert the physician of kidney injury, these alerts are usually only visible when allograft injury has already occurred. For example, serum creatinine does not detect subclinical TCMR (i.e., in the absence of graft functional deterioration), found in up to 30% of protocol biopsies [29]. For that reason, the histological analysis conducted on biopsies is still the gold-standard monitoring technique to evaluate allograft rejections processes as subclinical acute rejections [30,31]. As an early diagnosis, based on histological evaluation, will lead to more efficient treatments improving long-term outcomes, several groups have implemented protocol biopsies [30,31]. However, histological observation also presents considerable limitations, such as high costs, the risks that come with using such an invasive method, and possible sampling error of biopsies, not to mention the uncertainty in the identification of the rejection mechanism. For example, arteritis lesions due to TCMR lesions can also be present in ABMR, while ABMR diagnosis based on subtle microcirculation changes is difficult to identify and can be unspecific or unrelated to ABMR [13,17,20,25,32].

The development of sensitive and specific biomarkers to detect early events of rejection processes, such as subclinical rejection (SubCli-R) processes, while identifying the rejection mechanism would also enable more efficient and less toxic immunosuppressive therapies. These new biomarkers, besides allowing to start these therapies earlier, will enable adjusting of the immunosuppressive drug concentrations to a level that, while effectively blocking the rejection process, would minimize the toxic effects associated with these drugs (e.g., nephrotoxicity, cardiac diseases, diabetes, atherosclerosis, bone disease and infections such as chronic viral infections and malignancies) [33,34]. An ideal new biomarker would therefore enable the continuous monitoring of the allograft physiological state, allow personalized and efficient pharmacological therapy, and adjust drug types and dosages to each individual patient while simultaneously reducing the risk of immune rejection and the drug’s inherent toxic effects, ultimately leading to a higher quality of life and consequently a higher level of patient and allograft survival.

With that goal in mind, diverse biomarkers present in biofluids have been pointed out, as reviewed, e.g., by Eikmans et al. [35], Erpicume et al. [36], and Chakraborty and Sarwal [35,36,37]. However, none of these biomarkers have, to date and to the best of our knowledge, been used in regular clinical environments. For example, there are diverse works pointing to the urinary chemokines CXC chemokine ligand (CXCL) family, namely CXCL9 and CXCL10, as biomarkers for inflammation and AR, in which its levels may rise before the serum creatinine increase, decrease after immunotherapy against rejection, and may even be associated with a decrease in renal allograft function [29,38]. However, after validation of these biomarkers in independent cohorts, the maximum prediction value based on CXCL10 had a modest prediction value of subclinical rejection (AUC of 0.69, sensitivity of 0.61, and specificity value of 0.72) [39]. Therefore, there is no single biomarker in clinical use that is specific enough in order to detect rejection processes, especially in the case of early events, and based on minimal or non-invasive methods. The beforementioned emphasize the critical need to discover new biomarkers that can lead to the detection, as early as possible, of allograft rejection processes and to identify the immunological mechanism of rejection while enabling to define and adjust immunotherapies. The ideal biomarker should be obtained by rapid, cost-effective modes, with high specificity and sensitivity, promoting more frequent monitoring, all the while minimizing or even avoiding altogether the use of regular biopsies (Figure 1). This type of biomarkers will pave the way to the optimization of immunosuppressive therapies, leading to long-term safer immunosuppressive therapies tailored to each patient’s characteristics and to dynamic changes along the patient life span, which will, in the end, minimize kidney dysfunction and irreversible kidney damage, promoting a desirable higher quality of life for the patient and increased chances of graft survival.

## 2. Why Proteomics?

In theory, the omics sciences present advantages in the discovery of biomarkers in the diagnosis or prognosis of a defined physiological state due to the high complexity and interrelationships of the biological processes, as in this case, e.g., associated with the rejection mechanisms, immunotherapies, and the high diversity of the patient’s physiological states [40]. Since the proteome reflects different gene expressions, alterations occurring at the transcriptional level (e.g., alternative splicing), protein post-translational modification, and protein degradation or release portray with higher accuracy the system phenotype in a specific environment and, therefore, presents advantages over genomics and transcriptomics. For example, Nakorchevsky et al. [11] conducted transcriptomics and proteomics analysis of the kidney allograft biopsies presenting varying degrees of interstitial fibrosis and tubular atrophy (IFTA). The researchers observed that the expression of more than 1400 proteins traced the progression from normal transplant biopsies to biopsies with a severe disease where, in an initial analysis, a low (0.13) Pearson correlation coefficient between gene expression and protein was obtained. These researchers were able to correlate both data (gene expression and protein) only after focusing on the analysis of RNA transcripts and proteins associated with metabolic pathways identified as relevant in disease progression.

Two examples of proteomic studies, including renal biopsies of transplanted patients, were conducted by Nakorchevsky et al. and Lin et al. [11,41]. The former evaluated 33 biopsies from individuals with CR and diverse degrees of IFTA, whereas the latter analyzed 19 biopsies to discriminate between CR and renal adenocarcinoma. Nakorchevsky et al. [11] observed 492 unique proteins related to the progression from normal to mild, to moderate, and severe disease, including proteins associated with immune responses, inflammatory cell activation, and apoptosis. Meanwhile, Lin et al. [41] observed that 87 out of 1587 identified proteins were 1.5-fold higher in CR in relation to adenocarcinoma. There are also diverse proteomics studies focusing on urine and plasma/serum proteome to predict kidney function [42], including in patients with cancer [43,44] and diabetes [45]. The high specificity and sensitivity of these biofluids proteomics analyses enabled the detection of metabolic features associated with specific populations [46]. Concerning kidney allograft characteristics, there are several works pointing, for example, to the effect of donor-recipient age [47]. The present work will focus on the discovery of biomarkers based on biofluid proteomics, with the goal of predicting rejection processes, allowing for a faster, more effective and personalized route of treatment for each patient.

The most frequent analytical techniques to retrieve the system proteome are based on mass spectrometry techniques, including surface-enhanced laser desorption/ionization (SELDI) with time-of-flight (TOF) mass spectrometry (SELDI-TOF MS) or based on matrix-assisted laser desorption/ionization (MALDI) TOF MS. To increase the dynamic range of proteins analyzed and reproducibility, these methods are usually associated with previous techniques to separate peptides/proteins, including liquid chromatography (LC), capillary electrophoresis (CE), the less-used 2D electrophoresis [48,49], or the more recent affinity-based techniques [50]. The present work will not focus on or compare these techniques, as they were previously reviewed [48,49,50].

## 3. Which Biofluid to Analyze?

To discover biomarkers to be used under high-frequency monitoring, the analysis should theoretically be applied to biofluids such as urine and blood that can be obtained by non-invasive or minimally invasive procedures, respectively. For that reason, the present review focuses on proteomics of these biofluids even though direct analysis of the graft biopsy could enable a better understanding of the biological process mechanism, i.e., the biofluid only indirectly reflects renal allograft processes.

With that in mind, serum, plasma, and peripheral blood mononuclear cells (PBMC) have been explored since these biofluids and cells generally reflect the organism’s pathophysiological state, while the immune cells of the PBMC pool will reflect in a more concise manner, the immune system status. The major challenges associated with these biofluids proteomics for biomarker discovery result from the fact that these biofluids only indirectly reflect the kidney allograft status. For example, Flechner et al. [51] observed a different gene expression pattern on activated PBMC from the lymphocytes present at the graft biopsy. Another limiting factor associated with these biofluids pertains to their wide diversity and range of protein concentrations (approximately as high as 10 orders of magnitude), difficulting the identification of the lowest abundant proteins, thus limiting the reliability of the target protein identification and quantification [52,53,54,55]. Due to that, Freue et al. [56] and Perez et al. [57] depleted the 14 most abundant plasma proteins before proteomics analysis. The sample processing and storage before analysis will also significantly affect the proteome [58,59,60]. For example, protein inactivation should be conducted as soon as possible after the sample collection to avoid proteolysis.

Urine, besides being collected by non-invasive modes, could more directly reflect the kidney allograft status. Nevertheless, some limitations can also be associated with urine samples. For instance, this biofluid can be more varied in composition (e.g., varying according to food and liquids intake) and more prone to bacterial contamination, therefore also susceptible to stability problems. Furthermore, proteolysis may occur in urine within the bladder [59]. Due to the selectivity of the kidney glomerular basement membrane in excluding proteins, usually, a lower protein concentration of approximately 1000-fold lower than in blood is observed [41]. This can have an impact on the ability of detection in relation to plasma proteins [61]. However, due to the high sensitivity of proteomics, even a functional normal kidney may result in hundreds of detected urine proteins [62]. For example, Swensen et al. [63] detected ~28,000 urine peptides spanning ~2200 unique proteins, where one-third were plasma membrane proteins and another third were extracellular proteins, with ~400 unique to the study. Indeed, the kidney transcriptomic points to the kidney expressing 68% of all human genomic proteins, in which around 400 presented elevated expression in this organ compared to other tissues [63,64,65]. In addition, to minimize the shield effect of the more abundant proteins, some researchers also conducted depletion of the most abundant urine proteins before proteomics [29]. Urine can, therefore, and despite its challenges, present a very suitable and interesting alternative to blood analysis (Figure 1).

As expected, some proteins can be present in both biofluids, i.e., in blood (serum or plasma) and urine, as with collagen and fibrinogen alpha peptides [66]. Despite this, most of the proteome is specific for each biofluid. He et al. [66] did not observe a statistical correlation between the 6278 and 1743 urine and plasma peptides (*p* = 0.11), whereas Magalhães et al. [67] observed only 90 peptides to be common between the 1461 urine and 561 plasma peptides. As expected, urine presents an increased proportion of low-molecular-weight protein and peptide components relative to plasma since urine has a “natural filtration” process in the kidney [41].

Independently of the biofluid chosen, a high standardization of the collection, storage, and sample processing should be conducted. For example, Schaub et al. [61] observed that storing non-centrifuged urine samples can result, after a freeze-thaw cycle, in intracellular proteins released from leukocytes, red blood cells, and epithelial cells, leading to an altered molecular profile. A high modification of plasma and PBMCs proteome and transcriptome was also observed as early as 6 h after blood was drawn due to diverse cellular activation processes across diverse cell types [68].

## 4. Biofluids Proteomics

### 4.1. Urinary and Blood Proteomics

A semi-narrative search for manuscripts focusing proteomics on discovering biomarkers to predict rejection processes was conducted. For that, it was searched on PubMed articles presenting on the title or on the abstract, the combination of keywords “proteomics” and “kidney” or “renal” or “transplant”. From all the manuscripts retrieved, attention was focused on works pointing to biofluids proteomics developing predictive models of the rejection process, as summarized in Table 1.

To develop robust predictable models, it is crucial to use properly characterized samples based on parallel histological analysis of kidney biopsies. For example, IFTA could result from both immune rejection processes and non-immune [69,70]. Therefore, only by analyzing the histological kidney biopsy is it possible to identify the underlying pathophysiological process. In addition, stable transplanted patients may present SubCli-R, i.e., histological rejection findings associated with non-impaired allograft function [71,72,73]. Table 1 highlights with an asterisk (*) the pathophysiological states that were validated by biopsy analysis.

Most of the studies were conducted on urine, probably due to its non-invasiveness nature and the expectation of better models due to its more direct association with kidney function in relation to blood. It was also observed that the peptide panel used to build the predictive models for a specific phenotype, e.g., AR, depends on the biofluid analyzed and the analytical technique applied to retrieve the proteome. The peptide panel, even based on similar analytical techniques and biofluid, also depends on the patient’s pathophysiological state. For example, both Heidari et al. [74] and Ho et al. [29] analyzed urine by LC-MS/MS. However, Heidari et al. [74] developed a model predicting ABMR based on the epidermal growth factor, collagen alpha-1 (VI) chain, and Nidogen-1, while Ho et al. [29] developed a model predicting SubCli-R based on matrix metalloproteinase-7. It is also observed that most of the studies focus on the prediction of AR, where only a few evaluate the prediction of acute TCMR [75], or acute ABMR [74], SubCli-R [71,76], and chronic ABMR [69,70].

There are diverse works based either on urine or blood analysis that resulted in very suitable prediction models of rejection processes. For example, Heidari et al. [74] detected, from urine proteomics, 1020 proteins, from which 20 were differentially expressed in patients with ABMR in relation to patients with a stable kidney allograft, where 3 proteins resulted in a suitable ABMR prediction (AUC = 0.95). While Sui et al. [77], based on serum proteomics, were able to predict 83% of AR and 99% of CR. A high limitation associated with both these studies is the low number of enrolled patients and population homogeneity. Heidari et al. [74] developed the models based on 22 patients with ABMR and 14 patients without rejection processes, whereas Sui et al. [77] based the study on 12 patients with AR, CR, and without rejection, respectively. Aiming to discover biomarkers reflecting clinical reality, some researchers included a higher dimension population with a higher diversity of phenotypes. For example, Sigdel et al. [78] considered 396 kidney transplant recipients presenting acute rejection, chronic allograft nephropathy, BK virus nephritis (BKVN), and stable graft. A final panel of 11 peptides enabled to predict AR (AUC = 0.93), 12 peptides the chronic allograft nephropathy (AUC = 0.99), and another 12 peptides the BKVN (AUC = 0.83).

The identification of the immune rejection mechanism (e.g., TCMR vs. ABMR) will potentiate proper immunotherapy but also its adjustment by identifying the period in which the rejection is resolved. For example, Freue et al. [56], based on proteomics, identified a set of proteins as biomarkers of AR used to evaluate the effect of the immunotherapy efficacy on resolving the rejection process.

**Table 1 proteomes-10-00024-t001:** Proteomics studies on kidney allograft rejection conducted on urine (first rows), plasma, or serum samples (last rows), pointing to the analytical technique used, the population phenotype, dimension, and the predictive model’s output. The studies were sequenced according to biofluid used (from the urine studies followed by blood studies) and year of publication (from the earliest to the oldest).

Biofluid Type Proteomic Technique	Population Dimension (It Is Indicated If an Independent Validation Set Was Used)	Prediction Models (Peptide Fragments/Proteins Used in the Model)	Ref
Urine (14 peptides previously discovered)	*No-A-TCMR 390, borderline A-TCMR 157, A-TCMR IA+B 36. A-TCMR IIA+IIB+ I 46 (3 countries)	AUC (A-TCMR) 0.67 (collagen a(I) and (III) chain fragments)	[75]
Urine LC-TOF MS/MS	*STA 14, A-ABMR 22 Validation set: *STA 18, A-ABMR 19., HC 12	AUC (A-ABMR) 0.95, sensitivity 1.00, specificity 0.78 (epidermal growth factor, collagen alpha-1 (VI) chain, Nidogen-1)	[74]
Urine iTRAQ LC-MS/MS	*STA 117, AR 112, CAN 116, BKVN 51 Validation set: *STA 47, AR 42, CAN 46, BKVN 16	AUC (AR) 0.93; AUC (CAN) 0.99; AUC (BKVN) 0.83 (AR: 11 peptides; CAN: 12 peptides; and BKVN: 12 peptides)	[79]
Urine LC-MS/MS	*STA 5, Sub-Cli-R 6, IFTA 6 Validation set: *STA 22, ScR 17, GN 15, Viral nephropathies 7, IFTA = 20, IFTAi 13; B-T 13	AUC (matrix metalloproteinase-7: creatinine, inflamed vs. non-inflamed biopsies) 0.74	[29]
Urine SELDI-TOF-MS	*STA 26, AR 26 Validation set: *STA 16, AR 16	AUCs (alpha-1-microglobulin) 0.81 and (haptoglobin) 0.76	[80]
Urine CE-MS	*STA 23, Subcli-TCMRC 16 Validation set: *STA 36, SubCli-R 18, Cli-R 10	AUC (TCMRC) 0.91 (collagen α (I); α (III); matrix metalloproteinase-8)	[71]
Urine SELDI-TOF-MS/protein chip array	*STA 36, AR 55, ATN 10	ATN vs. STA: sensitivity 1.0 and specificity 1.0; STA vs. AR: sensitivity 0.86 and specificity 0.85 (*p* < 0.001) (ATN vs. STA: 2655; 11,730; 13,134 Da. STA vs. AR: 2364; 33,344; 66,479 Da)	[81]
Urine LC-MS/MS and ELISA	*STA, AR 10, HC 20 Validation set: *STA 20, AR 20, HC 20	AUC (CD44) 0.97; AUC (PEDF) 0.93; AUC (UMOD) 0.85 (MHC antigens, complement cascade, extracellular matrix proteins)	[54]
Urine MALDI-TOF MS	*STA 10, AR 10, BKVN 6 Validation set: *STA 10, AR 10, BKVN 4, NS 10, HC 10	AUC (AR) 0.96 (40 peptides)	[82]
Urine LC-MALDI-TOF MS	*STA 8, C-ABMR 10, IFTA 8, HC 10 Validation set: *C-ABMR 8, IFTA 6	AUC (C-ABMR) 1.00 (6 peptides—m/z:1539.8, 1540.03, 1542.1, 1575.48, 1587.86, and 1657.4)	[69]
Urine LC-MALDI-TOF MS	*STA 5, C-ABMR 10, IFTA 8, HC 9 Validation set: *STA 9, C-ABMR 11, IFTA 10, HC 9	C-ABMR: sensitivities 0.70 and specificities 0.70 (m/z: 610.7, 638.0, 642,6, 645.6, and 1096.8)	[70]
Urine SELDI-TOF-MS/ Protein chip array	*STA 22, Sub-Cli-R 27 Validations set: *STA 14, SubCli-R 10	Sensitivity 0.90 and specificity 0.71 (m/z: 2761, 10762, 11729, 11940)	[76]
Urine SELDI-TOF-MS	*STA 22, AR 18, ATN 5, *dn*G 5, HC 28, UTI 5	AR vs. STA *p* < 0.0001 Detected (peaks I+II+III) 94% AR and 18% STA and 0% HC	[61]
Urine SELDI-TOF-MS	*STA 22, AR 23, HC 20	sensitivity 0.905–0.913 and specificity 0.772–0.833 (2003.0, 2802.6, 4756.3, 5872.4, 6990.6, 19,018.8, 25,665.7 Da)	[55]
Urine SELDI-TOF-MS	*STA 15, AR 17	Tree decision model: sensitivity 0.83 and specificity 1.00 (decision trees 3.4, 10.0 Kd)	[83]
Plasma LC-MS/MS	*STA 25, A-CR 6	*p* < 0.05 (24 proteins)	[57]
Plasma plus Blood iTRAQ MALDI-TOF/TOF MS/MS	*AR 20, non-AR 20	AUC (21 peptides) 0.57 AUC (90 probes gene) 0.71	[84]
Plasma iTRAQ MALDI-TOF/TOF	*AR 11, non-AR 21	AUC 0.86 (titin, kininogen-1, and lipopolysaccharide-binding protein)	[56]
Serum iTRAQ LC-ESI-MS/MS	*AR 3, HC 9	Q ≤ 0.05 (109 proteins)	[85]
Serum MALDI-TOF MS	*STA 12, AR 12, CR 12, HC 13	Identification 83% AR and 99% CR (AR: 18 peptides; CR: 6 peptides)	[77]

*, The classification was proven by biopsy analysis; MS—mass spectrometry; CE—capillary electrophoresis; LC—liquid chromatography; ESI—electrospray; iTRAQ—isobaric tags for relative and absolute quantitation; MALDI—matrix-assisted laser desorption/ionization; SELDI—surface-enhanced laser desorption/ionization; TOF MS—time-of-flight mass spectrometry; ABMR—antibody-mediated rejection; A-CR—acute cellular rejection; AR—acute rejection; ATN—acute tubular necrosis; BKVN—BK virus nephritis; B-T—borderline tubulitis; C-ABMR—chronic active antibody-mediated rejection; CAN—chronic allograft nephropathy; Cli-R—clinical rejection; CR—chronic rejection; GN—glomerulonephritis; HC—healthy non-transplanted controls; IFTA—interstitial fibrosis and tubular atrophy; IFTAi—IFTA and inflammation; NS—nephrotic syndrome; SubCli-R—subclinical rejection; SubCli-TCMR—subclinical T cell-mediated rejection; STA—stable renal allograft; TCMR—T cell-mediated rejection; UTI—urinary tract infection; AUC—area under the curve.

An ideal biomarker would also enable to predict rejection before functional renal impairment, i.e., before clinical observations of, e.g., serum creatinine levels increasing above a defined threshold. As an example of this type of work, Metzger et al. [71] analyzed CE-MS urine samples from 16 patients with subclinical acute T-cell-mediated tubulointerstitial rejection and 23 patients with a stable allograft. The researchers developed a model based on 14 peptides that, when tested against an independent data set (n = 64), i.e., with samples not used on model building, resulted in an AUC of 0.91, predicting 89% and 100% of SubCli-R and clinical rejections, respectively, and 78% of patients without rejection processes. Acute tubular injury in the biopsies or concomitant urinary tract infection did not interfere with these predictions. Unfortunately, these results were not validated in a multicenter study. Gwinner et al. [75] tested the 14-peptide panel pointed in Metzger et al. [71] against urine samples retrieved from 624 patients collected among 11 different transplant centers across 3 different countries (France, Germany, and Belgium). The urine samples were taken immediately before a biopsy within the first transplant year. Unfortunately, the model did not predict borderline TCMR, and predicted poorly acute TCMR (AUC = 0.60, sensitivity = 0.66, specificity = 0.47). Even after optimization of the peptide marker-set (focusing on seven peptides), the model’s predictive ability only slightly increased (AUC = 0.67), and in an independent patient cohort, increased the AUC value to 0.69. Most probably, these results reflect the limitations associated with the population used to build the original predictive models.

Considering all the above-mentioned observations, the following precautions are critical to designing future studies toward the promotion of robust predictive models:Define a specific proteomics technique since each technique will highlight a different set of proteins;Define a specific biofluid, as the proteomic is specific to the biofluid;High dimension of the population evaluated;Apply an independent validation data set to test prediction models;Consider a high diversity of confound conditions, including patients with and without rejection processes, including, e.g., kidney drug toxicity, ischemic/reperfusion injury, and infections, among other diseases;All samples should be classified according to a parallel and rigorous histological-based biopsy analysis;The prediction power of the model should be quantified by measures such as AUC, sensitivity, and specificity, among others.

### 4.2. PBMC Proteomics

From non-proteomic-based studies, it is known the association of PBMC-specific characteristics with the allograft rejection process, including, for example, the association between the increased number of cells producing IL-21 and rejection processes [86], cells containing specific microRNA, and early detection of chronic antibody-mediated rejection [87], PBMC expressing TIPE2 with CR [88], T cell maturation and immune senescence to risk stratification of infections and consequently of the organ rejection after transplantation [89], Th1 cells secreting IFN-γ and FoxP3 associated with AR [89,90], and PBMC expression level of FoxP3 to predict AR [91]. From proteomics-based works, it is noteworthy to highlight the work of Savaryn et al. [92], that evaluated the proteoforms of PBMC, through LC-MS/MS, from 40 samples (20 from AR and 20 with stable allografts, biopsy proved), observing 344 proteins and covering 2905 distinct protein isoforms (proteoforms), i.e., 344 proteins associated with 2905 post-translational modifications. The researchers identified 111 differentials abundant proteoforms between AR and stable allograft groups. This occurred with less than a 1% false discovery rate. This study stands out not only by having analyzed the PBMC proteome but also by evaluating proteoforms since most proteomics studies focus on protein and not so on its post-translation variations [92].

### 4.3. Exosomes Proteomics

Extracellular vesicles (EVs), including microvesicles and exosomes found in biofluids, can also be an excellent source of biomarkers since they may represent molecular patterns from the source cells and, consequently, the pathophysiological process. Indeed, these vesicles may affect signaling pathways in recipient cells, antigen presentation, modulation of apoptosis, and tissue regeneration [93,94,95,96,97]. Based on this, there are diverse works exploring the EV immunomodulatory activity to induce immune tolerance to the allograft. For example, Ezzelarab et al. [98] administered autologous regulatory dendritic cells pre-loaded with EVs from the donor peripheral blood mononuclear cells, while others injected EVs (i.e., without cells) obtained, e.g., from T-cells [99], dendritic cells [100] or from mesenchymal stromal cells [98,99,101,102,103,104].

EVs are so relevant in pathophysiological processes that their increased number can be used as a biomarker. For example, Tower et al. [105] observed that plasma C4d+/CD144+ EVs, in transplanted patients with ABMR processes were (n = 28), in average, 24-fold higher than in normal non-transplanted and healthy volunteers (n = 23, *p* = 0.008), and 11-fold higher than transplanted patients without rejection processes (n = 67, *p* = 0.002). Interestingly, patients receiving treatment against AMBR, showed a 72% decrease in EVs (n = 9, *p* = 0.01). Castellani et al. [106] also observed a significantly higher number of plasma EVs (*p* < 0.001) for patients with antibody- or cellular-mediated rejection processes in relation to transplanted patients without rejection processes. Castellani et al. [106] and Tower et al. [105] observed that the plasma EVs increase in number with rejection processes was also associated with the EVs’ decreased diameter. Despite the increased number of exosomes in immune rejection processes, to obtain highly specific and sensitive biomarkers, it is imperative to evaluate the exosome content. For example, Castellani et al. [106] developed suitable prediction models of ABMR and TCMR based on EVs with specific markers, with validation data of 86.5% accuracy, including surface markers CD3, CD2, ROR1, SSEA-4, HLA-I, CD41b, HLA-II, CD326, CD19, CD25, CD20, ROR1, SSEA-4, and HLA-I.

A major advantage of searching for biomarkers on exosomes instead of their corresponding biofluid is that EVs may present specific molecules in higher concentrations in relation to their corresponding biofluid, consequently leading to more robust biomarkers. For example, Sigdel et al. [107] identified nine proteins in urine exosomes that predicted AR. However, the same proteins in urine were so diluted that their analyses were not statistically significant in predicting AR. In addition, Alvarez et al. [108] did not find significant differences in neutrophil gelatinase-associated lipocalin (NGAL) between patient groups in the cellular fraction of the urine. In contrast, the NGAL in the exosomes fraction was significantly higher in patients that received a transplant from deceased donors and between patients with delayed graft function group vs. non-delayed graft function. EVs may also present proteins that are not encountered in the corresponding biofluid. For example, Sigdel et al. [107] and Bruschi et al. [109] observed that 20% and 28% of proteins found in urine exosomes were not present in urine, respectively.

EVs are a rich source for discovering protein-based biomarkers due to their rich protein content and protein diversity. For example, Pisitkun et al. [110] detected in renal transplant patients 1989 urine exosomal proteins, from which 353, 322, and 165 were specific to tubular injury, TCMR, and ABMR, respectively. These researchers also observed that 92% of the exosomal proteins were intrinsic to the exosomes, i.e., only a small fraction (8%) of the proteins were most probably proteins from urine that had adsorbed to exosomes. In addition, Gonzales et al. [111] observed that, from the 1132 proteins identified in urinary exosomes, even though a large number were integral membrane proteins involved in solute and water transport (i.e., predominantly representing apical transporters from renal tubule segments). Only approximately 16% (117 proteins) were associated with diseases as judged by their presence on the Online Mendelian Inheritance in the Man database, from which 34 are known to be associated with renal disease. Another feature that stood out for these researchers was the evaluation of phosphorylated proteins, with the detection of 14 phosphoproteins among 1132 proteins.

There are diverse works pointing to sets of EVs proteins that are significantly different during rejection processes, including the identification in plasma EVs of 17 proteins characteristics in patients with poor transplant outcomes (n = 10) according to their estimated glomerular filtration rate (eGFR) [112]. In urinary exosomes, Pisitkun et al. [110] observed 322 and 165 specific proteins associated with TCMR (n = 6) and ABMR (n = 6), respectively. Sigdel et al. [107] identified 11 proteins associated with AR (n = 10), while Lim et al. [113] observed that, from the 169 identified proteins, 46 showed an increase in the stable allograft group (n = 22) and 17 increased in TCMR group (n = 25). Two out of the 17 proteins (tetraspanin-1 and hemopexin) predicted TCMR (AUC = 0.74). It is worth highlighting the Jung et al. [114] research, conducted with urinary EVs, based on LC-MS/MS proteomics and used western blot analysis to confirm protein levels of biomarker candidates. In this study, 385 kidney transplant recipients were considered, from a cross-sectional multicenter study, including biopsy-proven chronic active ABMR (n = 26), long-term graft survival (n = 57), and rejection-free matched kidney transplant recipients (n = 10). In this study, two proteins, apolipoprotein A1 and transthyretin (APOA1 and TTR), discriminated chronic active ABMR from long-term graft survival (AUC = 0.93). Another protein, zinc-alpha-2-glycoprotein (AZGP1) discriminated chronic ABMR patients from the matched control group (AUC = 0.95, sensitivity = 0.85, specificity = 0.80, positive predictive value = 0.77, and negative predictive value = 1.00).

### 4.4. Multi-Omics Approach

In the long run, it is expected that the integration of multiple omics data sets generated from the same patients will lead to a better understanding of the molecular and clinical features associated with kidney allograft rejection and, consequently, will potentiate the identification of phenotypic biomarkers with applicability in the clinical scenario. In the oncology field, this strategy has proven to be effective. As an example, Chari et al. [115] in breast cancer and Mariani1 et al. [116] in colorectal cancer compared diverse omics to understand the biological process, pointing out that multi-omics could enable the identification of biomarkers based on a reduced number of samples, in relation to the use of unique omics. Furthermore, the use of multi-omics platforms can enable better predictive models. For example, Mariianil et al. [115] developed poor predictive models of cancer aggressiveness based on proteomics (AUC = 0.68), which was significantly improved by adding genes and microRNA data (AUC > 0.90). Despite that, few researchers have applied a multi-omics approach to transplanted kidney research. From these, it is worth highlighting the Ling et al. [82] work that integrated urine proteomics with biopsy and PBMC transcriptomic. These authors, based on 70 urine samples from 50 kidney transplant patients (20 AR, 20 stable transplanted patients, and 10 BKVN), and 20 non-transplanted patients (10 nonspecific proteinuria with native renal disease and 10 healthy age-matched volunteers), identified a panel of 40 urine peptides that, in an independent validation data set (n = 24), resulted in an excellent AR prediction (AUC > 0.96). The paired renal biopsy transcriptomic reveals alterations in biopsies in accordance with the urine proteomics, therefore validating the proteomic data. Furthermore, proteomics and transcriptomics data pointed to those changes in collagen remodeling, allowing for the characterization of AR and the corresponding proteolytic degradation products in urine as a non-invasive diagnosis approach.

## 5. Final Considerations

Proteomics enables insight into the complex pathophysiological processes associated with kidney allograft diseases, including graft rejection and, ultimately, graft loss. Consequently, proteomics presents characteristics that may lead to the identification of much-desired new biomarkers, which would be highly useful for the clinical management of those patients. This would be especially important if those biomarkers could be obtained in non-invasive or minimally invasive modes, such as from urine and/or plasma/serum. In the long run, this would enable more frequent patient monitoring, providing clinicians with more precise tools to identify early rejection events and immune rejection mechanisms, consequently enabling more rapid, secure, and efficient immunotherapies adjustments. Indeed, proteomics of these biofluids points to potential biomarkers that allow predicting the immune rejection processes of the renal allograft, including detection and discrimination of SubCli-R, acute, and chronic immune rejection process from other allograft nephropathies, and even identifying the immune mechanism of rejection.

Despite all the potential, most of the studies present critical constraints that hamper the biomarker translation to clinical practice. There is a general lack of standardized protocols for biofluids and their exosomes or cell collection, processing, and storage. This lack of standardization will critically impact the final pointed biomarker, leading to high variability between studies. Protein quantification in the case of biofluid as urine can present extra challenges. Most of the studies focus only on protein identification and not so much on protein quantification or evaluating proteins post-translational modification. Most studies are also based on low dimension populations, do not include the immune status or the rejection of immune mechanisms as based on biopsy analysis, and are reliant on a low diversity of etiological diseases leading to allograft dysfunction (including a high diversity of infections and non-immune-based nephropathies), lack longitudinal studies that would accompany the patients along with medication adjustment, and lack of independent data sets for validation and are missing multicentric studies, all of which have been an obstacle to the introduction of these approaches. In addition, the majority of studies focused only on a few numbers of proteins for subsequent validation, therefore not taking full advantage of the omics information potential.

It is therefore critical to develop standardized protocols for sample processing (e.g., whole blood samples), further evaluate proteoforms and protein quantification, to conduct better study designs while including a higher dimension population with a corresponding diversity of pathophysiological states. It is also paramount to evaluate patients throughout time and include independent and large-scale validation processes. In the first phases of biomarker discovery, it must be considered, besides a set of specific proteins, multi-proteins, and/or peptide patterns. The complementary use of other omics sciences, e.g., transcriptomics, could also support the biomarker validation process. These studies will enable more robust biomarkers, applicable to stringent patient monitoring protocols, to predict in timely fashion rejection processes, the early identification of the rejection mechanism, toward a more dynamic adjustment of the patient therapy, i.e., to more efficient and personalized immunotherapies, toward a higher quality of life for the patient and significantly higher graft survival.

## Figures and Tables

**Figure 1 proteomes-10-00024-f001:**
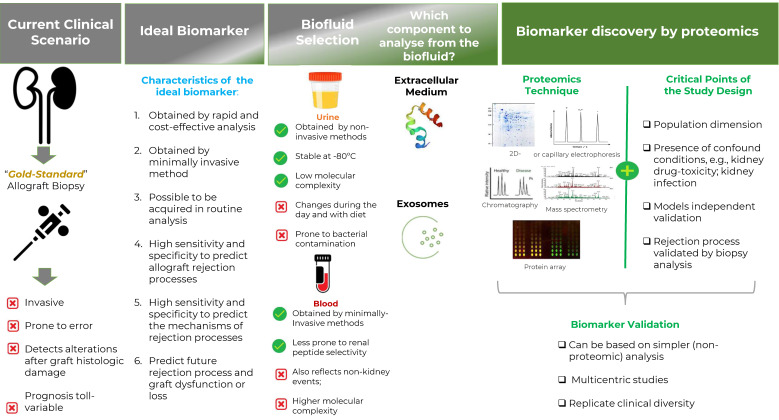
Main steps involved in the discovery and identification of biomarkers for diagnosis and prognosis of kidney transplantation rejection by proteomics.

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
