# Peer review of "Proteomics for Biomarker Discovery for Diagnosis and Prognosis of Kidney Transplantation Rejection"

_proteomes, 2022, doi:10.3390/proteomes10030024_

Round 1

Reviewer 1 Report

1. The Material & Methods section is lacking in the article. Therefore, I was not specified whether the article is a systematic review, or only a narrative review. I guess the latter is the case, but it should be specified.

2. Among the shortcomings of urine as a material for proteomic studies, risk of proteolysis in the bladder was not mentioned. I think it is an important issue that impacts the protocol of sample collection; the first morning urine is most probably not optimal for proteomics due to this reason. This should be added to the text.

Author Response

We thank the reviewer comments, that helped to improve the manuscript.

A semi-narrative search for manuscripts focusing proteomics to discover biomarkers to predict rejection processes was conducted. For that, a search on PubMed for articles presenting on the title or on the abstract, the combination of key words “proteomics” and “kidney” or “renal” or “transplant”, was conducted. From all the manuscripts retrieved, attention was focused on works pointing biofluids proteomics developing predictive models of rejection process. This explanation was included in the manuscript.

The risk of proteolysis in the bladder was now included in the revision version of the manuscript.

Reviewer 2 Report

This manuscript provides a literature survey of proteomic studies that have been conducted to identify biomarkers that are indicative for kidney rejection after transplantation. The review is well-written, logically structured, and up to date, and provides many details while not losing the main thread. The table is a useful resource providing details of recent studies, and their outcome. An additional asset is the summarizing statements they provide with regard to what can be learned from those previous efforts, while articulating recommendations for the future. Since many of the described studies focus on proteomic analysis of body fluids, lessons-learned will also be of interest to biomarker discovery in disease areas beyond kidney rejection. The manuscript has some minor glitches in grammar/spelling, e.g. biomarker (omit plural) in the title, ‘developed a poor predictive models’ (line 140), ‘samples whole processing’ (line 184, please re-word), so this may be checked throughout the text. Other than that, I recommend the paper to be accepted for publication.

Author Response

We thank the reviewer analysis and comments regarding the manuscript. We also apologize for the English mistakes. Thank you very much.